

**Climate change is rapidly deteriorating the climatic signal in Svalbard glaciers**
Andrea Spolaor[1,2], Federico Scoto[3], Catherine Larose[4], Elena Barbaro[1,2], Francois Burgay[5,2], Mats P.
Bjorkman[6], David Cappelletti[7], Federico Dallo[2], Fabrizio de Blasi[1,2], Dmitry Divine[8], Giuliano
Dreossi[1,2], Jacopo Gabrieli[1,2], Elisabetta Isaksson[8], Jack Kohler[8], Tonu Martma[9], Louise S. Schmidt[10],
Thomas V. Schuler[10], Barbara Stenni[2], Clara Turetta[1,2], Bartłomiej Luks[11], Mathieu Casado[12] and
Jean-Charles Gallet[8].
[1]CNR-Institute of Polar Science (ISP), Campus Scientifico, Via Torino 155, 30172, Venice-Mestre, Italy.
[2]Department of Environmental Sciences, Informatics and Statistics, Ca' Foscari University, Venice, Italy
[3]Institute of Atmospheric Sciences and Climate, ISAC-CNR. S.P Lecce-Monteroni km1.2, 73100 Lecce, Italy
[4]Environmental Microbial Genomics, Laboratoire Ampère, CNRS, University of Lyon, France
[5]Paul Scherrer Institute, Laboratory of Environmental Chemistry (LUC), 5232 Villigen PSI, Switzerland
[6]University of Gothenburg, Department of Earth Sciences, Box 460, 40530 Göteborg, Sweden
[7]Dipartimento di Chimica, Biologia e Biotecnologie, Università degli Studi di Perugia, 06123 Perugia, Italy
[8]Norwegian Polar Institute, Tromsø NO-9296, Norway
[9]Department of Geology, Tallinn University of Technology, Ehitajate tee 5, 19086 Tallinn, Estonia
[10]University of Oslo, Department of Geosciences, Oslo, Norway
[11]Institute of Geophysics, Polish Academy of Sciences, Księcia Janusza 64, 01-452 Warsaw, Poland
[12]Laboratoire des Sciences du Climat et de l'Environnement, CEA–CNRS–UVSQ–Paris-Saclay–IPSL, Gif-
sur-Yvette, France
Corresponding author: andrea.spolaor@cnr.it

















**Abstract**

The Svalbard archipelago is particularly sensitive to climate change due to the relatively low altitude of its main ice fields and its geographical location in the higher North Atlantic, where the effect of the Arctic Amplification is more significant. The largest temperature increases have been observed during winter, but increasing summer temperatures, above the melting point, have led to increased glacier melt. Here, we evaluate the impact of this increased melt on the preservation of the oxygen isotope signal ($\delta^{18}O$) in firn records. $\delta^{18}O$ is commonly used as proxy for past atmospheric temperature reconstructions and, when preserved, it is a crucial parameter to date and align ice cores. By comparing four different firn cores collected in 2012, 2015, 2017 and 2019 at the top of the Holtedahlfonna ice field (1100 m. a.s.l.), we show a progressive deterioration of the isotope signal and we link its degradation to the increased occurrence and intensity of melt events. Although the $\delta^{18}O$ signal still reflects the interannual temperature trend, more frequent melting events may in the future affect the interpretation of the isotopic signal, compromising the use of Svalbard ice cores. Our findings highlight the impact and the speed at which Arctic Amplification is affecting Svalbard's cryosphere.

**Introduction**

Arctic regions are undergoing faster warming than the global average, due to the so-called "Arctic Amplification" (Dahlke et al., 2020). Arctic Amplification is caused by various feedback processes in the atmosphere-ocean-ice system and significantly affects the Arctic North Atlantic region. Arctic warming is not seasonally uniform and has the largest impact in the winter months and close to the surface (Rantanen et al., 2022a; Dahlke and Maturilli, 2017). Furthermore, it is not evenly distributed across the Arctic; the largest warming rates are over the Barents/Kara Seas, where autumn and winter sea-ice retreat is strongest (Lind et al., 2018; Isaksen et al., 2022, 2016). However, even at tropospheric levels, there is a significant warming signal in recent decades that peaks in the Svalbard region, and more generally, in the North Atlantic sector of the Arctic (Dahlke and Maturilli, 2017). Rates there are up to four times the global average since 1979 (Rantanen et al., 2022b).

Glaciers and ice caps in the Svalbard archipelago cover an area of ~34,000 km$^2$, representing about 6% of the world's glacier area outside the Greenland and Antarctic ice sheets. Svalbard glaciers contain $7740 \pm 1940$ km$^3$ (or Gigaton; Gt) of ice, sufficient to raise global sea level by $1.7 \pm 0.5$ cm if totally melted (Schuler et al., 2020; Geyman et al., 2022; van Pelt et al., 2019) and are experiencing among the fastest warming on Earth (Noël et al., 2020) as a result of Arctic Amplification, and being situated at the edge of retreating Arctic sea ice.



Ongoing climate trends also affect the state of the seasonal snowpack in Svalbard (Østby et al., 2017;
van Pelt et al., 2016). For example the numbers of days with snow-cover on the ground has decreased
from 253 (1976-1997) to 219 (2006-2018). The change in the Svalbard climate has strong
repercussions for the entire environment of the archipelago. For example, there has been an increase
in frequency of Rain on Snow (RoS) events (Wickström et al., 2020) which lead to pervasive ice
layers(Sobota et al., 2020) covering the ground, limiting access to food for reindeers (Peeters et al.,
2019). The reduction in sea ice is limiting and changing the hunting area of polar bears. From a
climate perspective, the transformation from one regime to another is gradual and requires centuries,
as demonstrated by paleoclimatic studies, such as ice core investigations. Ice cores contain
information about past climate conditions and atmospheric composition including traces of natural
events (such as volcanic eruptions), past temperature reconstructions (Wolff et al., 2010) and
anthropogenic contamination(Vecchiato et al., 2020). Such studies revealed that rapid climate
changes have occurred in the past. For example, during the last glacial period in the Arctic, the so-
called Dansgaard-Oeschger events occurred when temperatures rose by about 5°C (Boers, 2018).
However, even during these natural abrupt events, a complete transition from stadial (glacial) to
interstadial (warm) conditions took about a century (Scoto et al., 2022; Steffensen et al., 2008).
Current temperature rise in Svalbard is much faster than the one observed during the D-O events,
with annual mean surface air temperature increasing in average by +1.3 K ± 0.7 K per decade, and
winter mean temperature increasing by +3.1 ± 2.4 K per decade (Dahlke et al., 2020; Maturilli et al.,

93 2013).

(Vance et al., 2016; Spolaor et al., 2016). Snowmelt and water percolation at the sampling site can
move the chemical constituents across the layers (Spolaor et al., 2021; Avak et al., 2019) disturbing
the original signal. Prolonged events can even fully compromise the preservation of the climatic
information contained by ice cores. Avak et. al. (2019) showed that atmospheric composition was
well preserved in an Alpine ice core during the winter, but that the melting in the spring and early
summer caused a preferential loss of certain major ions and trace elements. In particular, the elution
behavior of major ions is most likely controlled by redistribution processes occurring during snow
metamorphism, as underlined by recent work investigating the distribution of impurities within the
ice matrix (Bohleber et al., 2021). Variable mobility has also been observed for trace elements,
although they have been suggested to be better preserved than major ions. The Antarctic and the
Greenland plateau are the best locations for such studies, since the temperature is below the melting
point (<0°C), although rare melting events occur in the Greenland plateau(Bonne et al., 2015).
However, ice cores retrieved from these locations do not provide more regional climatic information.
To overcome this limitation, many other drilling sites have been investigated, for example in the Alps





(Arienzo et al., 2021; Gabrielli et al., 2016; Schwikowski et al., 1999) the Himalayas (Thompson et
al., 2018; Dahe et al., 2000), the Andes mountain range (Hoffmann et al., 2003), the Canadian
Arctic(Zdanowicz et al., 2018) and the Svalbard archipelago (Isaksson et al., 2005; Wendl et al.,

111    2015).

There are several ice caps in Svalbard, but given their relatively low altitude, most are not suitable
for the preservation of a pristine climate archive. The glacier equilibrium line altitude (ELA) varies
across the different regions of the archipelago but is generally situated between 300 to 700 m a.s.l.(van
Pelt et al., 2019). In the southern part of the archipelago, the ELA is lower due to the higher winter
snow accumulation, while in the northern part, the ELA rises to 600-700 m. Signal preservation
requires drilling to be above the ELA, for regular snow accumulation, but also, so that summer
percolation only moderately affects the upper firn layers.
Several drilling operations have collected ice core records in the archipelago, in particular in the
northern part. The longest (in time coverage) ice-core record was collected from Lomonosovfonna,
at 1230 m a.s.l., and covered ~1200 years of Svalbard climate history (Divine et al., 2011). Other ice
cores have been collected from Austfonna (750 m. a.s.l.), covering approximately 900 years
(Watanabe et al., 2001), Vestfonna (600 m a.s.l.) covering approximately 500 years (Matoba et al.,
2002) and Holtedahlfonna (1140 m a.s.l.) covering approximately 300 years (Beaudon et al., 2013).
To evaluate the robustness of Svalbard ice cores for future climate studies, we analysed the oxygen
isotopic composition ($\delta^{18}$O) of a sequence of four shallow ice cores collected at the top of the
Holtedahlfonna ice field in different years, each covering an overlapping atmospheric deposition
period, to provide a view of the evolution of isotopic stratigraphy over time. We focalize our study
on the $\delta^{18}$O since is the parameter most used in the ice core science for reconstruct the past
temperature change (Divine et al., 2011; Stenni et al., 2017) and it is one of the less affected, compare
to the other chemical parameters analyzed in ice core, by the melting and percolation events (Pohjola
et al., 2002). Several studies dealing with different elements and compounds have already been
performed using shallow cores from the summit of the Holtedahlfonna ice field, demonstrating the
importance of the site for climate studies (Burgay et al., 2021; Barbaro et al., 2017; Spolaor et al.,
2013a; Ruppel et al., 2017).  Results were linked to glacier mass balance measurements and snowpack
modelling. Based on our results, we observed that the climate signal is progressively deteriorating,
although the long term (>5 years) climate variation still seems preserved. This underscores the
urgency for obtaining records to help understand the climate processes occurring in one of the fastest
changing environments on Earth.

**2. Methodology**



### 2.1 The Holtedahlfonna ice field


Holtedahlfonna (HDF – Figure 1) is the largest ice field (ca. 300 km$^2$) in northwestern Spitsbergen,
located about 40 km from the Ny-Ålesund research station. It covers an elevation range of 0–1241 m
a.s.l. (Nuth et al., 2017) and the upper part of the glacier, located approximately at 1100 m a.s.l., has
a positive annual snow mass balance, ca. +0.50 m. w.e. a$^{-1}$(Beaudon et al., 2013; van Pelt et al., 2019).
The site has already been studied for long term paleoclimate reconstruction, covering the past 300
years (Divine et al., 2011; Goto-Azuma et al., 1995). In April 2005, a 125 m a long ice core was
drilled using an electromechanical corer and the bottom temperature in the borehole was –3.3°C,
assuring cold ice conditions over the entire ice thickness. Ice temperature measured in the borehole
featured a maximum of –0.4°C at 15 m depth, indicative of firn-warming due to the release of latent
heat from refreezing (Beaudon et al., 2013).
### 2.2 The Holtedahlfonna shallow firn cores: collection and processing
In the spring seasons of 2012, 2015, 2017 and 2019, a total of four shallow cores were obtained from
the summit of the Holtedahlfonna ice field (79°09′N, 13°23′E; 1150 m. a.s.l.). The shallow cores were
collected using a 4″ fiberglass Kovacs Mark-II ice corer driller powered by an electric drill and
reached depths of 7-10 m into the firn. All shallow cores were drilled from the bottom of the annual
snowpack\last summer surface. Length and density of each firn core section were logged, stored in
plastic sleeves, and transported back to Ny-Ålesund for laboratory analysis. For cores collected in
2012, 2017 and 2019, core samples were processed in a class-100 laminar flow hood in the laboratory
of the Italian research station "*Dirigibile Italia*" in Ny-Ålesund. Core sections were cut into pieces of
5 to 7 cm length using a ceramic knife and the external part of the core physically removed to avoid
contamination. The density was measured for each sample produced. The core 2015 was processed
as reported in Ruppel et al. (2017).
### 2.3 Oxygen stable isotope analysis ($\delta^{18}$O)
The samples for oxygen isotopic analyses ($\delta^{18}$O) were melted at room temperature (≈20°C) and
transferred into 2-mL clear glass vials filled to the top. Samples were kept refrigerated at +4°C and
analyzed at Ca Foscari University of Venice (2017 and 2019) and at Tallinn University of Technology
(2012 and 2015). In both cases, the isotopic measurements were carried out using a Picarro L1102-*i*
analyser coupled with a CTC Pal autosampler. The instrument uses Cavity Ring-Down Spectroscopy
(CRDS) technology, based on the unique near-infrared absorption spectrum of each gas-phase
molecule. The autosampler injects the melted sample into the vaporizer (set at 110°C), where it
becomes gaseous and is then transferred into the cavity (nitrogen is used as a carrier), in which the
measurement occurs. The instrument datasheet reports an analytical precision of ± 0.10 δ‰ for $\delta^{18}$O.



Each sample was injected eight times: only results within ± σ from the 8-repetition average were kept
for records, while outliers were discarded. Internal isotopic standards periodically calibrated against
IAEA-certified standards (V SMOW 2 and SLAP 2) were used for calibration.

**181    2.4 Holtedahlfonna surface mass balance**

Surface mass balance (SMB) of Holtedahlfonna is monitored by the Norwegian Polar Institute
(Kohler, 2013). SMB is obtained from repeated field visits at the end of winters and summers, with
winter snow-depth sounding and density measurements and repeated height readings of an array of
stakes along the glacier centerline. Balance estimates are extrapolated over the entire glacier basin by
determining the balance as function of elevation and averaging them, applying weights determined
from the distribution of glacier area as a function of elevation. This method quantifies the glacier-
wide SMB, i.e., the mass changes at the surface of the glacier, and within near-surface layers, but
does not include internal mass changes below the last summer surface. SMB measurements at
Holtedahlfonna started in 2003; since the drilling site is in the accumulation area, these measurements
provide information of the seasonal accumulation, but disregard the internal accumulation that may
occur due to refreezing of meltwater in layers below the last summer surface. The uppermost part of
the Holtedahlfonna (HDF) has had a consistently positive mass balance and is therefore assumed to
preserves most of its annual snow deposition.

**196    2.5 Estimation of Meteorological condition at the summit of the Holtedahlfonna ice field**

In absence of in-situ meteorological measurements at the drill site, we obtained long-term seasonal
(DJF, MAM, JJA and SON) temperature and precipitation series from the high-resolution CARRA
dataset (Copernicus Arctic Regional Re-Analysis, Schyberg et al., 2020). This 2.5 km resolution
product covering the period 1991-2020 is downscaled from ERA5(Hersbach et al., 2020) using the
state-of-the-art weather prediction model HARMONIE-AROME (Bengtsson et al., 2017). CARRA
has several improvements compared to ERA5, including assimilation of a large amount of additional
surface observations, extensive use of satellite data, and improved representation of sea ice; it is
therefore likely to provide the best estimate of meteorological conditions in the Barents Sea region.
The CARRA reanalysis is also used to force the CryoGrid community model  (Westermann et al.,
2023) to simulate glacier mass balance, seasonal snowpack evolution and meltwater runoff across
Svalbard Franz-Joseph Land and Novaya Zemlya. The model couples the surface energy balance and
a multi-layer subsurface module to resolve meltwater production, percolation, storage, refreezing and
runoff, accounting for the interaction with local density and temperature stratigraphers. The vertical


discretization comprises 47 layers of variable vertical extend to cover the uppermost 20 m below the
surface (Steffensen Schmidt et al., 2023).

**3.RESULTS**
**3.1 Shallow firn core dating and alignment**
To date the core, we use the seasonal cycle (where present) of the $\delta^{18}O$ data together with the mass
balance data available since 2003. Core depths were converted to water equivalent using the density
data acquired during the core processing. Density for the 2015 core is taken from Ruppel et al., (2017),
the 2012 values are published in Spolaor et al., (2013b), and density for the 2017 and 2019 cores are
presented in this work; density profiles of the four shallow cores (Figure S1) all reveal a similar
pattern.
The cores were collected within 50 m of the mass balance stake HDF-10. The stake measurements,
which show a consistently net positive mass balance, provide a historical record of snowpack
accumulation that can be directly used to assign a specific year to firn core depth range (Figure 2).
Oxygen stable isotopes ($\delta^{18}O$) can be used independently to annually date the ice, but only in ice-
core archives where the seasonal signal is well preserved. This means that snow accumulation needs
to be sufficiently high, and the summer ablation should not compromise the stratigraphy by
redistributing and smoothing the original atmospheric signal. By combining the annual accumulation
and the core depth expressed in water equivalent and the seasonality of $\delta^{18}O$ (where available and
preserved), we can date and align all four cores (Figure 3).
The cores cover 14 years (2004 to 2018). The time coverage for each core is reported in Table 1
together with additional information for each firn core. The 2012 core had a $\delta^{18}O$ average value of -
15.3 ± 1.0 ‰, the 2015 core a value of -15.1 ± 0.8 ‰, the 2017 core an average value of -14.4 ± 0.7
‰ and the 2019 core an average value -14.1 ± 1.2 ‰. The cores have a good overlap (Figure 3 and
6), and show a general increasing trend in $\delta^{18}O$ from 2004 until 2018. In particular, the 2012 and
2015 cores have similar trends, particularly during 2005-2006, a feature also useful for core
alignment. They also showed similar trends in the remaining periods that they each covered, though
with minor differences. The high values in $\delta^{18}O$ determined in year 2013 in the 2015 core are also
clearly found in the 2017 core, helping to synchronize the records. The alignment of the 2019 core
with previous cores could only be done through mass balance values, since the $\delta^{18}O$ values did not
show the same peaks as the other records. In particular, the decrease in $\delta^{18}O$ values recorded in the
period representing 2016 was not present in the 2017 core.

**3.2 Meteorological condition at the Holtedahlfonna ice field summit**



The meteorological conditions at the Holtedahlfonna ice field summit from 1991 to 2020 were
retrieved from model re-analysis and provide a clear overview of the on-going changes occurring at
the site.
The annual average winter temperatures (DJF) at the HDF summit (located at 1100 m a.s.l.) ranged
from -25°C to -15°C, and show an increasing trend of 2.37°C per decade for the period 1991–2020
(Figure 4a - blue line). The annual average spring and summer temperatures (MAM) ranged from -
17°C to -12°C (Figure 4a - green line) and -5°C to -1°C (Figure 4a - red line), respectively. The
average temperature increase per decade since 1991 was 0.38°C for spring and 0.51°C for summer.
The temperature during fall (SON) increased by 1.47 °C per decade and ranged from -15°C and -5°C
(Figure 4a - brown line).
Although the average seasonal summer temperatures were below the water melting point, positive
degree days (PDD – Figure 4b, expressed as the sum of mean daily temperatures for all days during
a period where the temperature is above 0°C), occurred at the summit of HDF, causing snowpack
melting. The cumulative annual PDD, retrieved from model temperature series outputs, showed a
stable value for the period 1990 to 2015, although some years (1994, 1999, 2010) and periods (2001–
2006) were characterized by an increased PDD. A net increase from 2015 to the present time was
recorded. Snow melting at the site was clearly visible and confirmed by the presence of several ice
lenses in the core (Spolaor et al., 2013b; Burgay et al., 2021).
The annual model estimated precipitation (1991–2020) ranged between 630 to 1170 mm w.e. per
year, with a slight increase in the most recent period (Figure 4d). A similar trend was also observed
in Ny-Ålesund (Førland et al., 2020). Seasonal precipitation (Figure S2) was most abundant during
fall (SON) and winter (DJF), with an average precipitation of 286 mm w.e. and 274 mm w.e,
respectively, and a relative average contribution of 32% and 31%, respectively, to the total deposition.
The lowest precipitation occurred in spring (MAM) and summer, with an average precipitation of
170 mm w.e. and 145 mm w.e., respectively, which represents an average contribution of 20% of the
total deposition in spring and 17% of the total precipitation in summer.
Although the annual mass balance was always positive, the summer mass balance was both positive
and negative depending on the meteorological conditions (Figure 2). The winter accumulation
represented between 60% and 100% of the net annual mass balance at the site. Even though the
summer mass balance data from 2015 to 2020 were positive, melting also occurred and water
percolated into the snow and firn before refreezing.
Most of the melting occurred during the summer period (JJA), but melting events also occurred during
fall and late spring (Figure S3). The estimated annual melting at the site from 1991-2020 (Figure 4c)
varied between 960 mm w.e (2020) and 117 mm w.e (2008) and showed a clear increasing tendency



following temperature rise. Moreover, autumn snowpack melting events, previously rare, became a
more regular phenomenon in the period 2015 to 2019. However, spring snowmelt is sporadic (2011)
and rare.
In addition to meteorological reanalysis from the HARMONIE-AROME model, the CryoGrid
simulation provided information about the presence of liquid water in the firn and its penetration
(Figure 5). Percolation was mainly confined to the surface layer between 1991 (beginning of the
simulation) to the end of the 90s(except 1999). Percolation increased significantly from 2000
onwards. In particular, for the period 2004-2005, severe surface melt events occurred (Figure 2c and
Figure S3), causing water percolation for several meters (Figure 5). The 2006 to 2014 period was
characterized by relatively limited surface melting and the lowest amount of percolated water, which
did not exceed one (2006 and 2008) to four (2010 and 2011) annual snow accumulation periods.
Based on the model's calculations, water percolation increased since 2014 and was able to reach
deeper firn strata. Although the model suggests the presence of liquid water in the firn, water and
elution channels are complex to simulate and likely present high spatial variability. Hence, we only
consider the data presented in Figure 5 in a qualitative manner to evaluate the possible presence or
absence of liquid water within the snowpack and its theoretical penetration\percolation depth.

## 4. DISCUSSION

The aim of the paper is to evaluate the effect of temperature rise on the $\delta^{18}$O Holtedahlfonna ice core
signal preservation. Our discussion will focus only on the periods covered by the shallow cores.
Based on the $\delta^{18}$O records of the four shallow cores, it is evident that the seasonal signal experienced
considerable changes and progressively deteriorated in the most recent cores. The most important
parameters affecting the pristine atmospheric signal trapped in the snow is the amount of snow
melting, which depends on the snow and meteorological conditions, and the penetration of the melt
water into the snowpack.
In the core collected in 2012 (Figure 3), the seasonal variations are clear for almost the entire period
except for 2004-2005, a period characterized by significant summer melt that disturbed the
atmospheric signal trapped in the ice. However, for the period 2006-2011, the seasonality is clear and
each $\delta^{18}$O seasonal cycle is confined within the annual snow mass balance measurements.
The 2015 core still presented the seasonal cycle in the upper half of the core, corresponding to the
second period (2010-2014). However, the seasonal feature of the $\delta^{18}$O identified in the core 2012 for
the periods 2008–2009 was no longer present, suggesting a possible elution caused by the percolation
of liquid water (Figure 5). The model simulation supports the possibility that post deposition events
may have occurred within the firn due to the percolation of liquid water.



The most striking change in terms of the $\delta^{18}O$ seasonal cycle occurred in the 2017 core. The 2017
core overlapped with the 2015 core for the period 2012-2014 and, while the seasonality for this period
was well defined in the 2015 core, only the seasonal $\delta^{18}O$ for year 2013 was visible in the 2017 core.
The $\delta^{18}O$ seasonal cycle of 2014 has undergone significant smoothing and the $\delta^{18}O$ seasonal cycle in
2012 is no longer visible. For the period 2015-2016, the seasonal cycle was not clear, although
oscillations were still present.
In the most recent core collected in 2019, a seasonal $\delta^{18}O$ cycle could no longer be detected and
particular features, such as the drop in the $\delta^{18}O$ signal in 2016 (not observed in the 2017 core), was
not linked to a drop in the temperature, since 2016 was the warmest year on record (Figure 6, red
dots).
Two independent statistical analyses, one using the significant value of a regression model and the
other using the spectral analysis, were performed on the shallow core records to test the presence of
seasonal oscillation on the $\delta^{18}O$ signal. Both statistical analyses demonstrated the disappearance of
the seasonal signal in the most recent (2017 and 2019) shallow cores (full details are reported in the
supplementary material - section 2).
The change in seasonality and, to a lesser extent, in the total amount of precipitation, might have
influenced the $\delta^{18}O$ signal of the four cores. However, from the model results, the seasonal
contribution to the total annual precipitation did not change significantly (Figure S2). This would
suggest that precipitation does not play a central role in explaining the degradation, or possible
change, in the $\delta^{18}O$ signal, and that increased melting and water percolation might have had a larger
effect. Instead, the increase in year-round precipitation could enhance melt water formation during
the summer periods. The preservation of the ice core climate signal strongly depends on the amount
of snow melt during summer and the capability of water to penetrate the snowpack, which in turn is
controlled by snow temperature. The progressive atmospheric warming, the increase of summer
melting and water percolation as well as the water movement within the snowpack could all have had
an impact on the $\delta^{18}O$ signal present in the Holtedahlfonna firn\ice.
The progressive degradation and loss of the seasonality of the $\delta^{18}O$ signal in the shallow core (2004-
2018) is also supported by the results obtained from the $\delta^{18}O$ signal in the 2005 core. In the deep core
collected in 2005, the seasonal signal of the $\delta^{18}O$ in the period 1960 to 2000 was well preserved
(Figure S5). The signal determined in the 2005 Holtedahlfonna deep ice core shared similar features
with those determined in the 2012 and 2015 shallow cores, where the seasonal oscillations were still
partially present, but not with signals determined in the 2017 and 2019 cores, where the seasonality
in $\delta^{18}O$ almost disappeared. We suggest that since 2015, estimated melting and percolation increased



because of the evolution of the general atmospheric conditions, causing a deterioration of the climate
signal preserved in the firn\ice.
Water stable isotopes are commonly used as a temperature proxy. By overlapping the water stable
isotope profiles measured in the shallow cores and, comparing their trends with the annual average
temperature, we suggest that the general atmospheric temperature trend is still preserved within the
HDF ice (Figure 6), although some clear deterioration is visible. For example, the highest annual
temperature values recorded in 2016 were not mirrored in the $\delta^{18}$O record from the 2017 and 2019
cores. This underscores the impact of high temperatures on the preservation of pristine atmospheric
signals in ice cores that have significantly impacted the preservation of the atmospheric signal, since
temperature values.

**5. Conclusion**
An ice core drilled at the summit of Holtedahlfonna has previously been used to provide atmospheric
and climate conditions about the past 300 years (Beaudon et al., 2013). Before 2005, the site was
characterized by moderate summer melting, but the snow and ice was shown to preserve important
climate information as well as the main seasonal features. The current warming of the Svalbard
archipelago has clearly enhanced glacial mass loss, with a rise in the equilibrium line altitude and a
shorter snow season. This study is the first investigating the impact of temperature rise on climate
signal preservation within the firn\ice in one of the highest ice fields in Svalbard. The direct effect of
higher temperatures has increased summer melt and enhanced meltwater percolation. In this study,
we have shown that the climate signal preserved in the ice has been progressively deteriorated. For
example, in seven years, the seasonal signal visible in the 2012 core has completely disappeared in
the 2019 core, most likely due to increased snow summer melting and water percolation. However,
although the $\delta^{18}$O seasonal signal has disappeared, the overall atmospheric warming signature is still
preserved in the ice\firn, suggesting that the site is still suitable for long record paleoclimate
reconstruction. However, with the current warming rate of the Svalbard archipelago and the
consequent increase in summer melting, Holtedahlfonna and other ice fields at similar altitudes might
no longer provide suitable records of the climatic condition. Glaciers worldwide are currently not
only losing mass at unprecedented rates, but also the climatic information they contain.



**Acknowledgments**
This work has been supported by the "Programma di Ricerca in Artico" (PRA, project number
PRA2019-0011, Sentinel); by the Svalbard Science Forum/Research Council of Norway through the
Arctic Field Grant call (project ASIHAD, ISSICOS, BIOMASS), by French Polar Institute IPEV
(Institut Polaire Français Paul-Emile Victor) science funding (programs 399 and 1192) and the
Svalbard Strategic Grant (project C2S3, nr. 257636, SnowNet nr. 295779 and BC3D nr. 283466).
This project has received funding from the European Union's Horizon 2020 research and innovation
programme under grant agreement no. 689443 via project iCUPE (Integrative and Comprehensive
Understanding on Polar Environments). This research has been partially funded by the University of
Perugia Research Action no. 5 "Climate, Energy, and Mobility". Cryogrid simulations have been
supported by the Nansen Legacy project (Research Council of Norway grant 276730) and SIOS
infraNor (Research Council of Norway grant 269927).

**Author contribution**
AS, EB, FS, JCG, CL, MB, JG, FB, and DC conceived the experiment and collected the samples and
wrote the paper with the support of all co-authors; CT, TM, GD and BS analyze the samples; JK
provide the field mass balance data and contribute in data interpretation; LSS and TVS provide
the model data and atmospheric re-analysis; FdB and MC perform the statistical exercise and
contribute in data interpretation. BL and FD contribute to data interpretation. DD and EI
provide the data from previous ice core and contribute to data interpretation.

**Data availability**
The data will be available upon request to the corresponding author.

**Competing interests**
The authors declare that they have no conflict of interest.



**FIGURES**

**Figure 1.** Location of the drilling site (red star) within the Holtedahlfonna (HDF) ice field as compared to the Ny-Ålesund research village (NyA). Maps from https://toposvalbard.npolar.no (last access: 5ᵗʰ June 2023).

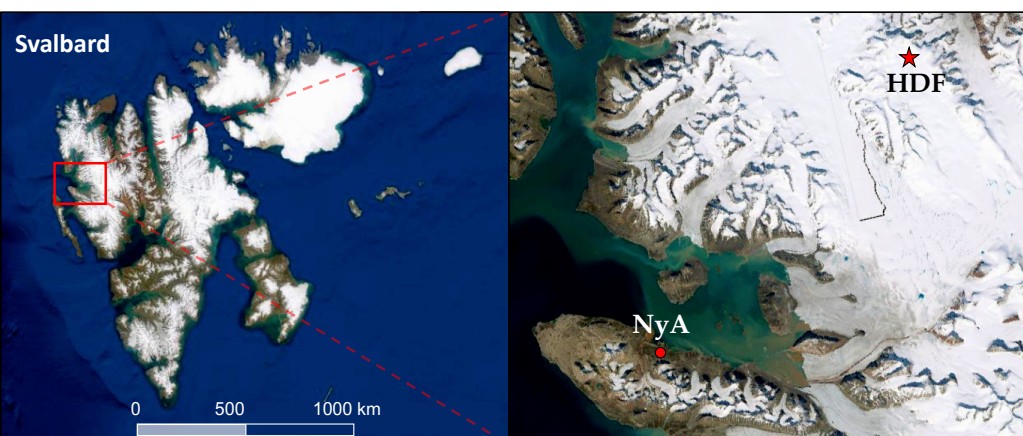



**Figure 2.** Mass balance measurements, modelled precipitation and snow melt at the drilling site. a) cumulative surface mass balance (SMB) expressed in cm of w.e., b) comparison of modeled total annual precipitation (green – in mm w.e) and modeled melt (red in mm w.e). c-e) net, winter and summer mass balance (cm w.e.) measured at the top of the Holtedahlfonna ice field, respectively.

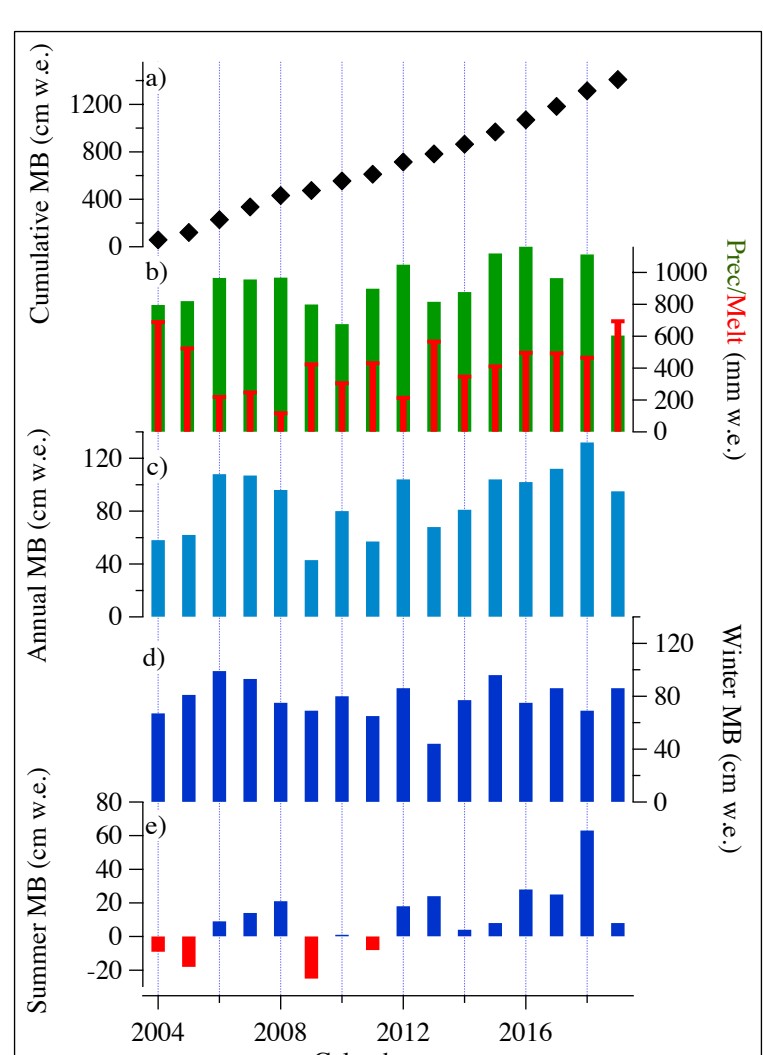

**Figure 3. Oxygen stable isotope profiles** $\delta^{18}O$ of the shallow cores. The shallow core was aligned by converting the depth to depth expressed in cm of w.e. using the annual mass balance (MB) data. The white and pink colors distinguish different years based on the MB measurements and are reported in the upper panel.

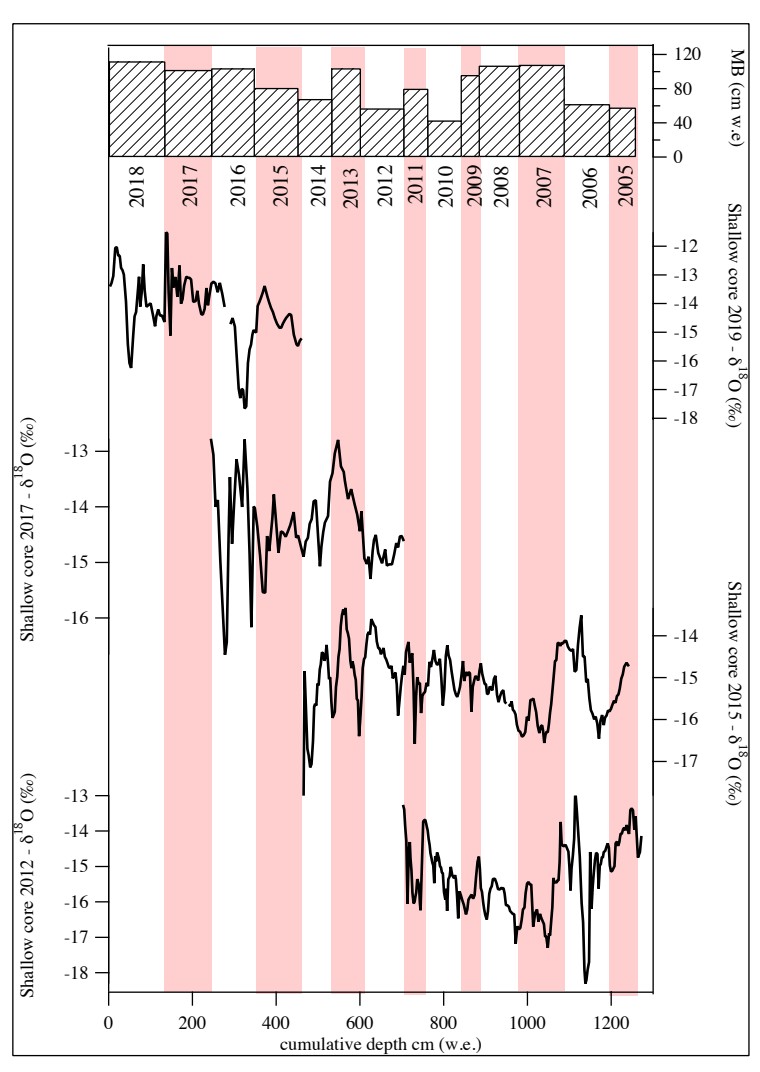



**Figure 4.** Modeled meteorological conditions at the Holtedahlfonna shallow core drilling site (1150 m a.s.l.) from 1991 to 2020 at seasonal resolution. a) winter (DJF - blue), spring (MAM – green), summer (JJA – red) and fall (SON – brown) temperatures, with increasing trend line for the period investigated. b) annual PDD value (grey). c) annual melting (in mm w.e in red). d) annual total precipitation (in mm w.e. – blue)

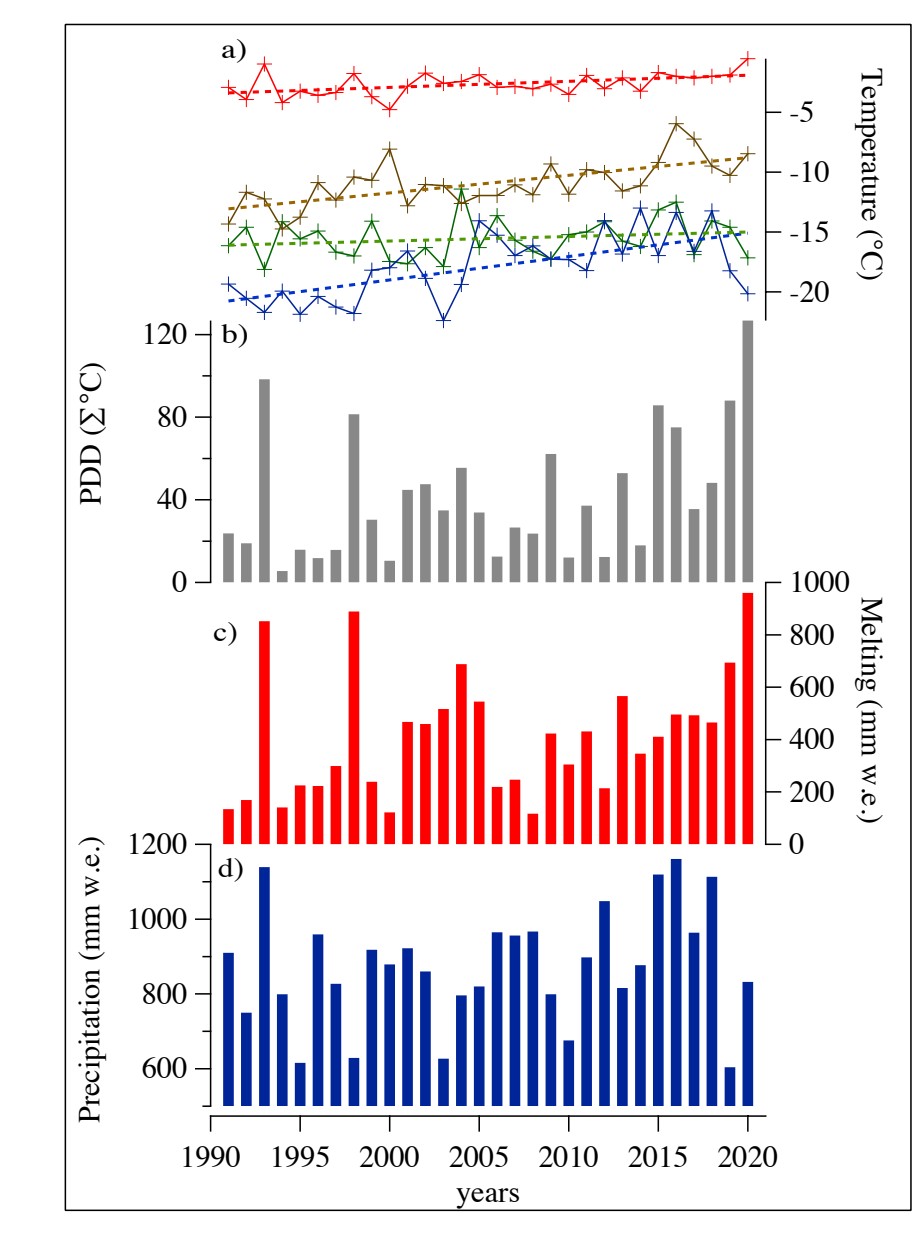





**Figure 5.** Evolution of the water content in the snowpack at the top of Holtedahlfonna estimated by model simulation between 1990 and 2020. The chart shows the volumetric water content (%) in the snow/firn (white to blue color), surface height evolution (black line), 0° C isotherm (red). Dashed lines show the period covered by the four shallow cores.

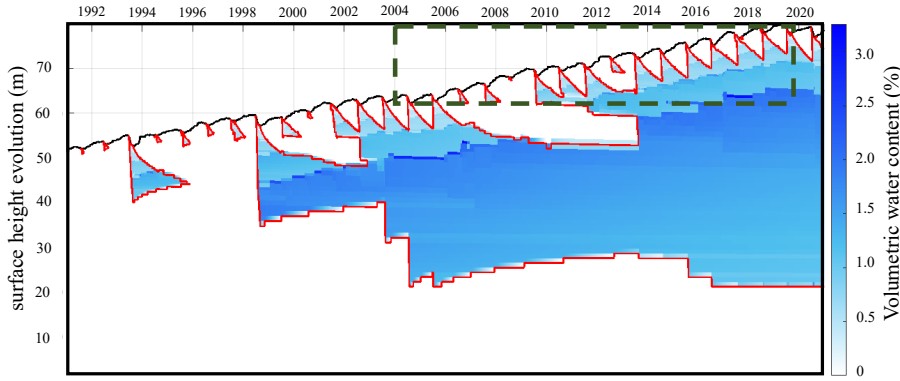





**Figure 6.** Estimated annual average temperature at the top of Holtedahlfonna ice field (red dots)
and the δ¹⁸O signal of the four shallow cores.

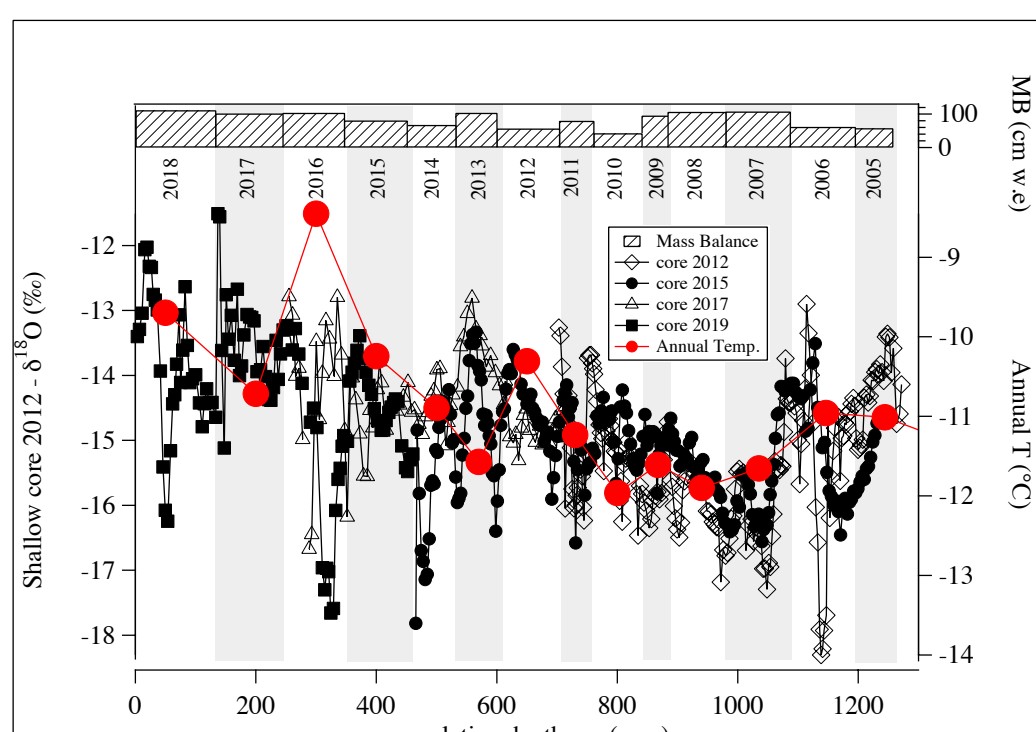



**TABLES**

**Table 1**. Shallow ice core descriptions. The table reports the length expressed in cm and in water equivalent (w.e.) and the estimated (Est. start year\Est. end year) time coverage.  The average density of the cores is also reported.

| Core ID | Length (cm) | Length (cm w.e.) | Ave density (kgL$^{-1}$) | Est. Start year | Est. End year | Drilling period | Reference |
|---|---|---|---|---|---|---|---|
| 2019 | 769 | 461 | 0.60 | 2018 | 2012 | April 2019 | This work |
| 2017 | 736 | 466 | 0.63 | 2016 | 2010 | April 2017 | *Burgay et a. 2017* |
| 2015 | 1185 | 832 | 0.70 | 2014 | 2005 | May 2015 | *Ruppel et al. 2017* |
| 2012 | 954 | 575 | 0.60 | 2011 | 2004 | April 2012 | *Spolaor et al. 2013* |





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
