# Peer review of "Andrea Spolaor1,2, Federico Scoto3,2, Catherine Larose4, Elena Barbaro1,2, François Burgay5,2, Mats P. Bjorkman6, David Cappelletti7, Federico Dallo2, Fabrizio de Blasi1,2, Dmitry Divine8"

_The Cryosphere, 2023_

## Author Response (AR1)

**Review 1**

The authors will assess the repercussions of this escalated thawing on the preservation of the oxygen isotope signature (δ18O) within firn records. Through a comparison of four distinct firn cores retrieved in 2012, 2015, 2017, and 2019 from the upper region of the Holtedahlfonna ice field (at an elevation of 1100 m a.s.l.), they observe a gradual deterioration of the isotopic signal. The authors attribute this decline to the augmented frequency and intensity of melting events as the effect of the Arctic Amplification influencing the Svalbard cryosphere.

**General comments**

The structure of the manuscript is very good, the methodologies are well explained, and results and conclusions are clear.

But personally I find a lack in the interpretation that must be taken into account..

The authors compare the stable isotope records of four firn core with the surface glacier mass balance from a stake (from glacier mass balance monitoring network) close to the drilling site. The results will provide indication of the deterioration of the isotopic signal in the glacier due to the general climate warming related to the Arctic Amplification.

The mass balance data are used mainly to date the core in the most precise way. The mass balance itself has not be considered as a parameter influencing the isotopic signal.

In the discussion of the glacier mass balance at the stake close to the drilling site, used for annual accumulation do not take in account the wind transport/erosion effects of the surface, that can move large quantities of snow and change the annual accumulation rate. At the same time this effect can be important on the stable isotope records, because snow from lower altitude (with different isotope signal) can be added to the snowpack at the drilling site. At the same time wind blow erosion can subtract part of the annual snowpack. For the same reasons the stable isotope records can be affected by these effects. The accumulation of the snow blown by wind can add layers with different isotopic signals not related to the local seasonality, and the erosion can subtract snow layers affecting the seasonality. In both cases the lack in the seasonality of then stable isotope records cannot be related to the change in local temperatures.

*We appreciate the points raised during the review process, and we want to emphasize that while wind redistribution is a factor to consider in ice core research, we do not believe it plays a significant role in our specific case or when interpreting water stable isotope ice core signals*

*It's important to note that many of the sites chosen for ice core climate archive recovery are located at glacier summits or domes, where processes like wind erosion, snow redistribution, and transport are common and, in some cases, quite efficient. These factors are inherent to the formation of water stable isotope ice core signals. While they may have long-term effects (such as changes in the main wind patterns or atmospheric circulation, which are beyond the scope of our paper), over shorter time scales (a few decades), changes in wind regimes should not be sufficient to account for the loss of the climate signal observed at the Holthedalfonna summit.*

*Additionally, analysis of wind patterns in Ny-Ålesund does not indicate any significant shifts or changes in average wind velocities, as reported by Cisek et al. in 2017. We acknowledge that this analysis relies on model estimations, but it represents the best approximation available for our study site.*

*Furthermore, while wind redistribution can move snow, it primarily affects snow deposited at similar altitudes, which tends to have a similar water stable isotope fingerprint. It is highly improbable that snow deposited at lower elevations could be lifted and deposited at the summit of Holthedalfonna in quantities significant enough to completely degrade the climate signal preserved in the ice.*

*For these reasons, we maintain our position that wind redistribution does not significantly contribute to the deterioration of the observed climate signal in our study. We added a short paragraph in the main text at line 298-304 to address this.*

**In effect all the records showed (fig 3), there are not any regular correlation between the stable isotope records and the mean annual accumulation (MB), except some sporadic layers (i.e. 2013/2012).**

*We do not suggest a direct scientific correlation between changes in mass balance and water stable isotopes. This expectation is grounded in the understanding that mass balance is influenced by a multitude of environmental factors and processes that differ from those governing water stable isotopes. These two aspects of glacial research are influenced by distinct sets of variables and mechanisms.*

**Clearly, the increase of the temperatures, the increase of the PDD's and melting are clear and the effect of the snowpack in the last years is in the direction of the deterioration of the climatic records, but must be taken in account all the meteorological parameters can affect the results.**

*We have taken into consideration various potential factors, and our analysis suggests that the primary driver behind the observed degradation is the increase in summer melting. While other factors, such as an increase in winter accumulation, could theoretically contribute by leading to more snowmelt during summer and increased percolation in the snowpack, confirming these effects would require dedicated modeling exercises and field experiments. However, the prevailing evidence points towards the heightened energy available in Svalbard, which translates into higher temperatures and increased snow melting, as the principal factor explaining the degradation we have observed. This factor appears to have the most substantial influence on the changes we've observed in the ice core data.*

**I suggest the authors try to improve the manuscript taking in account the possible wind effects, both for the mass balance and stable isotope interpretation.**

**Specific comment**

**Fig. 2 and 5 present the x-axis, the years inverted respect the fig 3, 4 and 6. Understand the problem of the depth of the records, but personally I suggest to use one only direction to help the readability of the figures.**

*We agree with the referee and modified the figure 2 and 5*

**Review 2**

In this manuscript Spolaor and colleagues present a study of how melt events triggered by rising temperatures due to global warming may affect paleoclimatic reconstructions from ice cores. This is a very timely study, and although Spolaor and colleagues investigate this phenomenon in Svalbard, their findings will be applicable to other sites as well from West Antarctica and Greenland.

The manuscript addresses an important topic and presents interesting data. However, the discussion is underwhelming and underdeveloped (see my last major comment). The authors should expand the discussion and in particular discuss the importance and applicability of their findings to other locations. For these reasons, the manuscript in its current form is not ready for publication and I recommend major revisions.

**Major Comments:**

The introduction is reasonably well-structured and the research gap is hinted at. The last few paragraphs should be rewritten to clearly define the current state of knowledge, the research gap, and the advance of this study.

*We thank the reviewer for his\her positive comments and have rewritten the last two paragraphs as suggested.*

In Figure 3 the time axis is inverted because the main axis is the depth, which is fine. But Figure 4 should go from 1990 on the left to 2020 on the right, just as Figure 2 and Figure 5. The inverted time direction makes no sense here and is confusing.

*This was also suggested by referee 1 and we have modified the axes to assign the same time direction in all the figures.*

The density measurements of the 2017 core were performed at much lower resolution than the other cores (Figure S1). In lines 319 to 324 the 2017 core is described as being much smoother than the others. How much of this smoothing can be attributed to the lower density measurements? This should be discussed here.

*Density measurements in core 2017 are simply performed at different resolutions compared to the water stable isotopes. There is no direct consequence regarding the interpretation of the climate signal presented for the core 2017.*

This manuscript is about the effect of percolation on the d18O signal variability. Figures S6 and S7 are the main figures for this discussion and should be in the main text with a thorough discussion, not the supplementary.

*We moved the statistical analysis figures to the main text as suggested, including a short section about these two figures (Line 331 – 335).*

**Minor Comments:**

Line 75 – 76: Please indicate where (geographically) these snow cover numbers were registered, as I imagine it may be quite different in other areas of the island.

*We have included details as suggested*

**Line 80-82: This is a problematic sentence, as it is too vague. What about tipping points? What about Dansgaard-Oeschger events? I suggest to either remove it or greatly expand on this subject.**

*We agree with the review and the sentence has been removed*

**Line 83: Reference missing for volcanic events.**

*Reference has been included*

**Line 86: Rephrase. DO events didn't occur when temperature rose by 5 deg. It's the abrupt event that was defined that way.**

*The sentence has been rephrased*

**Line 105 – 106: Of course Greenland and Antarctic ice cores also provide regional climate information! There is no limitation here!**

*We removed the sentence to be more clear and avoid any misunderstanding*

**Line 241: That is incorrect. By eye, only the last core (2019) appears to show a statistically significant positive trend. The other three appear to show periodic variability but no trend. Even the trend in the 2019 core may just be a section of a stable periodic variability.**

*We modified the sentence by substituting the word "trend" with "fluctuations with shared features" that could be more appropriate to our discussion.*

**Line 244: 2012 core, not 2013.**

*We clarified the sentence: "The high $\delta^{18}O$ values in 2013 that occur in the 2015 core are also clearly found in the 2017 core"*

**Line 244: The 2012 and 2017 cores don't overlap. I think you mean that specific peaks in the three cores can be found in one of the other two. Rephrase**

*The sentence has been rephrased*

**Line 283 – 284: Maybe add a trend line in Figure 4c.**

*We thank the review for the suggestion but we prefer to keep the figure as it is to avoid making it overly confusing*

**Line 356, Figure 6: I assume the temperature data (black squares) are from the reanalysis? Please indicate the source of these data in the figure caption.**

*The black square is from the ice core data. However, to be clearer, the figure has been modify using different colors for each core. The red dots represent annual average temperature available from the Svalbard and Jan Mayen monitoring website.*

**Review 3**

The destruction of the climate records derived from ice cores caused by rising tropospheric temperatures has been observed at high and lower latitudes in the Northern and Southern Hemispheres. This paper discusses the degradation of the seasonal $\delta^{18}O$ variations in firn cores from the Holtedahlfonna (HDF) ice field in Svalbard caused by rising temperature over recent years. According to the authors, the compilation of the $\delta^{18}O$ records from four firn cores drilled between 2012 and 2019 cover the period from 2004 to 2019. (I have some confusion about the borders of that time period which will be addressed below.) The impacts of rising temperatures in the Arctic, aside from the serious and large-scale problems it will create for the ecosystems at these latitudes and for the global-scale climate, will result in the compromise of climate records from Arctic ice cores.

In my opinion it's important that glacier melt and the destruction of climate records from ice cores be studied and documented in as many places as possible. The data from the monitoring of this Svalbard glacier is a contribution, although the period of monitoring is relatively short (2005 to 2019) and assumptions are made about short-term changes in the seasonality of the $\delta^{18}O$ profiles that may not be supported as discussed below.

The authors provide a detailed analysis of the effects of rising Arctic temperature on the HDF firn cores in relation to local meteorological records and conditions at the drill site. [Note: Unfortunately, there was an oversight in the preparation of the manuscript that resulted in the exclusion of Figure 2 which is supposed to show annual and seasonal mass balance data; Figure 4 was accidentally dropped into the Figure 2 spot.] I find Fig. 6 to be a compelling summation of the authors' work as it shows clear similarities between general trends in temperature and $\delta^{18}O$ although I am intrigued by the lack of δ18O response to the very high 2016 annual temperature, which is noted by the authors although the opposing temperature/$\delta^{18}O$ trends in 2012 and 2013 are not mentioned. The 2015/16 El Niño was a major event that caused (up to that point) record warmth in many Arctic regions. Strong El Niños like the 2015/16 event leave isotopic signatures in glaciers in the Andes and the Tibetan Plateau. The 2005 HDF ice core record (Fig. S5) shows strong $^{18}O$ enrichment that is probably linked to the 1997/98 El Niño, although the 1982/83 event barely registers. Melt at HDF in 1998 was high (Fig. 5; Fig. S3), although the record does not extend back to 1982 so that comparison cannot be made.

*We appreciate the review's insightful points, and we understand that some of the issues raised are challenging to address with the data presented in our manuscript. The primary purpose of the data we have presented is to demonstrate a gradual and consistent degradation of the seasonality in the water stable isotopic signal.*

*While the El Niño events in 1998 or 1983 may indeed be related to the strong enrichment indicated by the reviewer, delving into the specific atmospheric processes behind such enrichments is beyond the current scope of our manuscript. Our main objective with this paper is to highlight the impact of Arctic warming on the preservation of the isotopic signal.*

*We want to emphasize that another paper is currently in preparation, which will focus more on the comparison between the 2005 record and the shallow core recovered and presented in this manuscript. However, combining both discussions into a single paper would result in a lengthy and dense document. Therefore, we have made the decision to split the work into two separate papers: one addressing the degradation of the signal (the current manuscript) and a follow-up paper that will delve deeper into the climate record and the processes influencing it. This division will allow us to provide a more comprehensive and detailed analysis of the different aspects of our research.*

**I have several suggestions which may improve the clarity of the MS:**

**1). Obviously, please include Figure 2.**

*We have included figure 2, apologies for the mistake.*

**2). Line 157: How deep were the snow pits in which the firn cores were drilled? If the cores were drilled in pits, why is the top of the 2019 core shown at the surface (0 cm) in Figs. 3 and S6?**

*In section 2.2, we clarify that "the shallow cores were collected using a 4-inch fiberglass Kovacs Mark-II ice corer drill powered by an electric drill. These cores reached depths ranging from 7 to 10 meters into the firn. Importantly, all shallow cores were drilled from the bottom of the annual snowpack or the last summer surface". This means that the surface or the starting point for the 2019 core, which we designated as 0 cm, was selected at the interface between the annual snowpack of 2020 and the firn from the summer of 2019. This specific horizon serves as our reference "0 cm" point, from which all other cores (2017,2015 and 2012) were aligned using the mass balance data and the depth expressed in cm of water equivalent.*

**3). The authors state in Lines 240-241 that "All the cores have an overlap period and show a general increasing trend in $\delta^{18}O$ from 2004 to 2018 (Figure 3)." Although the $\delta^{18}O$ averages are provided for each of the short records, it would be useful to the reader if that trend could be illustrated by compiling the 4 short records into one complete record (shown by time) and including a trend line. Since the records overlap it should be possible to match the isotopic features. In addition, it would be useful to the reader to place the trends of these short records into a longer context. The authors show a $\delta^{18}O$ record that extends back to 1960 from a core drilled in 2005 (Fig. S5); it would be interesting to see a compiled shallow firn-core $\delta^{18}O$ record appended to this earlier record. Do the authors have the data from the 2005 core up to 2005?**

*Regarding the first point raised about compiling the four short records into one complete record shown over time, we appreciate your perspective. However, although we understand your preference to present all the data from the four shallow cores in one single plot, we prefer to keep it as it is now. As the records presented cover a limited time span, it makes sense to align the cores using depth in millimeters or centimeters of water equivalent. This approach allows for a more detailed presentation of the data without the need for age-scale conversion, which could potentially reduce data and introduce artifacts. Regarding the second point and the comparison between the 2005 core and the shallow cores presented in this manuscript, we understand your point but this will be a topic for another manuscript that is more closely related to climate dynamics and ice core science. Moreover the main goal of this manuscript is indeed to show the differences in the isotopic records on the overlapping periods among the 4 shallow cores: since these differences are clearly identified, it means that putting together in a single record the different isotopic records could lead to errors.*

**4). I am confused as to why the authors assert that the $\delta^{18}O$ record extends back to 2004. Fig. 3 shows that the earliest core (2012) only extends back to 2005, with perhaps only a couple of cm occurring in 2004.**

*We agree with the reviewer and have modified the text accordingly.*

**5). Lines 243-244: This sentence is confusing: "The high values of $\delta^{18}O$ found in the 2013 and in the 2015 core are also clearly found in the 2017 core...". The way it's stated sounds like**

**there's a 2013 core, which there is not, so I assume what is meant is "The high values of δ¹⁸O in 2013 that occur in the 2015 core are also clearly found in the 2017 core…".**

*We modified the sentence as suggested to "the high δ¹⁸O values in 2013 that occur in the 2015 core are also clearly found in the 2017 core"*

**6) Lines 314-318: This may be a premature conclusion. There are post-depositional processes other than melt and percolation that can affect the isotopic signal. For example, the authors do not include any information on seasonal wind strength at HFD, and although the annual snow mass balance (0.5 m w.e.) is high, strong wind events can redistribute a great deal of surface snow which can disrupt the seasonal isotopic variations. Since winter accumulation accounts for most of the net mass balance and most of the melting occurs in summer, is it possible that winds in winter can redistribute surface snow before melting occurs?**

*The same point has been raised by reviewer 1. We copy below the reply given to review#1:*

*We appreciate the points raised during the review process, and we want to emphasize that while wind redistribution is a factor to consider in ice core research, we do not believe it plays a significant role in our specific case. It's important to note that many of the sites chosen for ice core climate archive recovery are located at glacier summits or domes, where processes like wind erosion, snow redistribution, and transport are common and, in some cases, quite efficient. These factors are inherent to the formation of water stable isotope ice core signals. While they may have long-term effects (such as changes in the main wind patterns or atmospheric circulation, which are beyond the scope of our paper), over shorter time scales (a few decades), changes in wind regimes should not be sufficient to account for the loss of the climate signal observed at the Holthedalfonna summit. Additionally, analysis of wind patterns in Ny-Ålesund does not indicate any significant shifts or changes in average wind velocities, as reported by Cisek et al. in 2017. We acknowledge that this analysis relies on Ny-Alesund data, but it represents the best approximation available for our study site.*

*Furthermore, while wind redistribution can move snow, it primarily affects snow deposited at similar altitudes, which tends to have a similar water stable isotope fingerprint. It is highly improbable that snow deposited at lower elevations could be lifted and deposited at the summit of Holthedalfonna in quantities significant enough to completely degrade the climate signal preserved in the ice.*

*For these reasons, we maintain our position that wind redistribution does not significantly contribute to the deterioration of the observed climate signal in our study. Anyway, we add a short paragraph in the main text at line 298-304*

**7) Fig. S1, density profiles: It would be helpful to see years on each of the profiles so that they can be more easily compared with the corresponding δ¹⁸O data. Alternatively, the authors could create a supplement figure that shows the density and δ¹⁸O data for each of the firn cores side by side in depth or in time.**

*We can understand the interest in showing the δ¹⁸O data in age scale, since this parameter is depending (mainly) on the climatic condition that changes year by year, and this is shown in figure 4. However, we do not fully understand why it would be necessary to show the density profile plotted in the age scale or together with the δ¹⁸O signal. The density increase depends on the firm densification process, which, in turn, almost only depends on the depth. We are aware that the firn densification and hence the density is site specific and the rapidity with which the density can increase depends on the site characteristics (meteorological and glaciological). Plotting together the density*

*profile of the 4 cores, or together with $\delta^{18}O$ signals might not bring additional information from our point of view. All the shallow cores have been collected from the annual snow pack\firn transition during different years. Considering that the densification of the firm did not change significantly we do not understand how to plot the density profile of the 4 cores all together. We might have not fully understood the question so we apologise in advance for this.*

**Small editing issues:**

**The MS should be corrected for numerous spacing mistakes between words. I suggest that the revised version be carefully proofread before submission.**

*We noticed these issues and have carefully checked the entire manuscript*

**Line 108: There are more recent papers on ice core climate records from the Andes (e.g. Vimeux et al., 2009 Palaeogeography, Palaeoclimatology, Palaeoecology 281, 229-242; Thompson et al., 2021 Global and Planetary Change 203, https://doi.org/10.1016/j.gloplacha.2012.103538).**

*The reference list has been updated*

**Line 196: "HDF" was already defined at the beginning of the MS, it doesn't need to be redefined here.**

*We modified accordingly.*

**Line 229: "$\delta^{18}O$" is already used previously in the MS but it is defined here. It should be defined the first time it appears.**

*We modified accordingly.*

**Line 315: reverse "core" and "2012"**

*We modified accordingly.*

**Line 346: remove the negative sign before 2000**

*We modified accordingly.*

---

## Author Response (AR2)

**AUTHOR'S RESPONSE**

**Spolaor and colleagues have significantly improved their manuscript. Most of my comments and suggestions have been adequately addressed. However, the introduction still suffers from major structural issues:**

**The last two paragraphs have been changed, but not necessarily for the better. Although several previous studies are now cited from the same region (118-123), only the age range of the ice cores are mentioned. Although an interesting variable, this has not much relation with the current study. The idea behind the state-of-the-art paragraph is to give a short overview of previous results that can be compared to the present study. In this context, the state-of-the-art should summarize previous results and discussions relating to major ion mobility, trace element mobility, etc. Similarly, previous results in other regions of the globe should be summarized. For example, the authors state that "many other drilling sites have been investigated, including the Alps, the Himalayas, the Andes, Canada, and Svalbard" (106-110). This is great, but the authors should summarize here the results and discussions of the cited papers in relationship with the main subject of this manuscript. This will help identify the research gap, which is still not clearly defined in the introduction. Finally, the lines 128-139 read more like a discussion and this text should be merged with the discussion section. Instead, a final introduction paragraph (Here we …) should be written.**

**Here are some guides with information about each section:**
**https://www.scidev.net/global/practical-guides/how-do-i-write-a-scientific-paper/**

**https://researcheracademy.elsevier.com/writing-research/fundamentals-manuscript-preparation/structuring-article-correctly**

**The introduction is a very important section of any manuscript and needs to be improved. At this stage, I recommend minor revisions. Once the introduction has been improved, the manuscript will be ready for publication.**

We appreciate the thorough review of the manuscript. We have taken into careful consideration the comments and suggestions provided by the referee, and we have made revisions accordingly. In the new version of the manuscript, specific changes have been implemented in the introduction, now between lines 119 to 159. Notably, the text between lines 128-139 of the previous version has been modified, and lines 135 to 139 have been removed. We believe these revisions enhance the clarity and coherence of the introduction in response to the valuable input from the reviewer.

---

## Author Response (AR3)

**Dear Andrea, the paper certainly improved after the review process. However, there are still several points that need to be addressed before this manuscript can be published. Please be sure to make also a very extensive and careful check of the consistency of all the formats before submitting the next revised version.**

We have revised and improved all sections of the manuscript as recommended by the editor. Textual modifications are indicated in red, while changes made in the figures, although completed, are not highlighted.

**Abstract:**

**The abstract is rather vague and does not include a clear mention on important info contained in the main text such as 1) when the isotopic signal deterioration happened and 2) to which depth (year) its effects can be observed. Part of this info can be extracted from Line 376 "We suggest that since 2015, estimated melting and percolation increased because of the evolution of the general atmospheric conditions, causing a deterioration of the climate signal preserved in the firn\ice." and/or from Line 311 "311 Based on the model's calculations, water percolation increased since 2014 and was able to reach deeper firn strata".**

The abstract has been improved as suggested.

**Main Text:**

**Line 76: Please provide a clear reference for "data from Monitoring of Svalbard and Jan Mayen, mosj.no" according to the reference policy of this journal.**

Understood. This is an environmental monitoring system within the Government's environmental monitoring in Norway, not a journal. The data are thoroughly validated and fully accessible through the website without any specific references indicated for their use.

**Please use consistent unit of measurements of Temperature (C or K, not both; e.g. Line 87; 91-92) through the entire manuscript) and Time (e.g. Line 125 1600 - 1800 CE; Figure 2: X=year; Fig. S4 X= Date etc. ). Please use also consistent direction of time/depth in ALL the figures of the manuscript and supplementary info.**

We modify accordingly and all the x axis are now presented as year

**Line 136 "showed a strong local response of chemical species". Please clarify what is the strong local response.**

We modify as follow "the obtained record suggests that there are local influences affecting the studied chemical species"

**Line 147 "In light of the accelerated warming". Please provide a reference as acceleration of the warming is different than amplification and is a very debated topic.**

We modify the accelerated warming with Arctic Amplification

**Lines 158-162. Are these lines retained or cancelled?**

These lines have been cancelled

**Line 225: "extensive use of satellite data". Please be specific.**

We modify by removing "extensive"

**Line 245 (Figure 2). Figure 3? We correct accordingly**

**Line 261: "The alignment of the 2019 core with previous cores could only be done through mass balance values since the d18O values did not show the same peaks as the other records". This implies that maximum and minimum values cannot be identified at least in the 2019 core. However, these were used for the regression analyses. To me this is still the main problem that needs to be clarified (please see also below).**

At the reviewer's request, the regression analysis has been included in the main text as suggested. The statistical method's identification of maximum and minimum peaks, not aligning with the mass balance data for the 2019 and 2017 cores, serves as statistical evidence indicating the degradation of the climate signal. However, for the 2015 and 2012 cores, the peaks identified by the statistical method align quite well with the mass balance data. We have enhanced the method's description for improved clarity

**Line 285: "with a slight increase (precipitations) in the most recent period (Figure 4d)." This sentence is contradicted by the graph that show the last two years (including 2019) having less precipitations. Please note that when a decrease in precipitation is combined to an increase in melting percolation, this may lead to a stronger deterioration of the signal.**

The most recent shallow core was obtained in 2019, yet the firn and the data presented commence from the last summer surface, which approximately corresponds to September or October 2018. In response to the upcoming question, we found it beneficial to incorporate additional data even beyond the shallow core's coverage period. Nevertheless, we have updated all figures, retaining 2019 as the final year for displaying meteorological data.

**Line 298 "The estimated annual melting at the site from 1991-2020 (Figure 4c) varied between 960 mm w.e (2020) and 117 mm w.e (2008)" Are 2020 data relevant at all for this paper? Is not the 2019 core the most recent one? Please check carefully through the paper.**

Please refer to the previous answer

**Line 307: "In particular, for the period 2004-2005, severe surface melt events occurred (Figure 2c and Figure S3)" Please highlight in Fig 3 the period covered by the firn core records.**

We believe the editor is referring to Figure S3. In Figure S3, we will emphasize the period covered by the record by highlighting it with a light yellow rectangular shape.

**Line 320: "it is evident that the seasonal signal experienced considerable changes and progressively deteriorated in the most recent cores." Which cores? Please be specific.**

We modify as follow: "it is evident that the seasonal signal for the core 2019 and 2017 experienced considerable changes and progressively deteriorated

**Line 359 "The change in seasonality and, to a lesser extent, in the total amount of precipitation, might have influenced the d18O signal of the four cores.". The "change in seasonality" of the signal? This is an effect not a cause of deterioration. Please rephrase.**

We rephrase as follow "The change in the seasonal patterns of precipitation, and to a lesser degree, the overall quantity, could have influenced the $\delta^{18}O$ signal of the four cores"

**Line 430 : "The data will be available upon request to the corresponding author." I encourage the authors to put the data in a public data repository as the professional email address of the corresponding author is always temporary and data will not be accessible in the future. Please check also the policy of the journal.**

We have made modifications and plan to deposit the data in the Zenodo repository once the paper is accepted. The link will be included during the proofreading stage.

**Line 597 "Figure 6: Identification of the annual minimum and maximum values of $\delta18O$ (red and blue points) based on the annual mass balance dating…". The identification procedure of minimum and maxima is not clear and at this time seems inconsistent/arbitrary between different cores. Just to give one example: Panel C: the 2016 minimum would be most obviously the one at around 325 cm we depth. So in conclusion at this time I'm not convinced of this analysis and I would suggest to consider to delete entirely this min-max identification and the consequent regression analyses presented in Fig. 7.**

The description of the statistical method has been enhanced for improved clarity, as previously mentioned. This statistical analysis has been incorporated into the main text based on the referee's suggestion.

**Supplementary info:**

**In general the format of the figures presented in the SI need to be consistent with the ones in the main text, for unit of measurements, size of characters, colors (e.g. of the seasons, of the years). Please make sure that the characters are large enough to be readable (Fig. 5-6 and 7).**

We've updated the figures in the Supplementary Information. Figures 5 to 7 have been generated using a different graphing software.

**Paragraph 2.1 "(Errore. L'origine riferimento non è stata trovata.)." Please delete.**
Thanks to notice this, we remove it